# Beyond Isolation: Effects of Defense Combinations on Evasion and Privacy Risks in Machine Learning

**Maria Terenteva**
ISP RAS
MSU
m.terenteva@ispras.ru

**Kirill Lukianov**
ISP RAS
ISP RAS Research Center for Trusted Artificial Intelligence
MIPT
lukianov@ispras.ru

## Abstract

Modern machine learning systems are increasingly required to withstand multiple threat vectors simultaneously, including evasion attacks and data privacy attacks. Yet most defenses are designed and evaluated in isolation, so their behavior under joint deployment remains poorly understood.

In this work, we study how combining defenses changes both robustness and privacy: do defenses reinforce each other, or do they interfere and weaken the intended guarantees? We focus on representative evasion defenses implemented through post-batch training modifications (e.g., gradient regularization) and combine them with privacy-preserving training based on differential privacy.

We evaluate four threat scenarios: no attacks, only evasion attacks, only privacy attacks, and simultaneous evasion and privacy attacks, using standard architectures and benchmark image classification setups. In addition to empirical results, we use analytic reasoning to interpret the observed interactions through the lens of optimization.

Our results show that interactions are non-trivial. Some combinations yield complementary gains, improving overall security without disproportionate utility loss, while others introduce clear conflicts that reduce robustness or increase leakage. These findings indicate that multi-factor protection cannot be achieved by naïvely stacking defenses and motivate evaluation frameworks that explicitly account for defense interactions in adversarial and privacy-sensitive settings.

## 1 Introduction

Machine learning systems are increasingly deployed in domains where failures can have systemic consequences, including critical infrastructure, public services, and safety-relevant pipelines Entezari et al. (2023); Pozdnyakov et al. (2024); Badjie et al. (2024); Raihan (2023); Adeoye et al. (2025); Litvinov et al. (2026). Expectations regarding reliability, security, and trustworthiness are formalized through regulatory initiatives by national authorities and organizations such as the European Commission and the National Institute of Standards and Technology. These initiatives require assessing machine learning solutions against a broad spectrum of risks, including adversarial manipulation and unintended information disclosure. Evaluation of defenses must therefore consider complex operational settings with multiple simultaneous threat vectors.

Research has addressed individual aspects of trust in machine learning Thulasiram (2025); Lukianov & Yaskov (2025); Yakushev et al. (2025); Rechkemmer & Yin (2022), including robustness to attack examples Raff et al. (2025); Zhang et al. (2025), protection of sensitive data Pelekis et al. (2025), interpretability Sazonov et al. (2024); Jhanjhi (2025), and reliability under distributional shifts Lu et al. (2025); Lukianov & Yaskov (2025). However, many studies examine these challenges in isolation, optimizing defenses against a single threat class while assuming others are absent Namatevs et al. (2025); Schwarz et al. (2024). In practice, threat models combine multiple evasion capabilities, and mitigation strategies are deployed jointly, creating uncertainty regarding interactions and combined effectiveness.

Evasion and privacy attacks are central because they directly affect operational integrity and regulatory compliance. Evasion attacks manipulate inputs at inference time to induce incorrect predictions Muthalagu et al. (2025); Zhang et al. (2020a); Wang et al. (2023). Privacy attacks, such as membership inference and data reconstruction, extract information about training data Shaikhelislamov et al. (2024); Liu et al. (2021); Rigaki & Garcia (2023). Joint mitigation of these attacks is critical in real-world deployments.

Defense methods include differential privacy (DP) for privacy protection Ponomareva et al. (2023); Zhang et al. (2020b), which bounds individual data influence via noise addition and gradient clipping, and various robustness methods against evasion, including adversarial training, regularization, smoothing, and post-batch modifications of gradients or loss functions.

The interaction between privacy and evasion defenses remains underexplored. DP modifies gradient distributions, potentially affecting decision boundaries and robustness Zhang et al. (2019); Qiu et al. (2019), while robustness regularization can alter sensitivity relevant to privacy leakage Zhang et al. (2020b); Ponomareva et al. (2023). Evidence on combined effects is limited, particularly for post-batch defenses integrated with DP Szyller & Asokan (2023); Sazonov et al. (2025); Thakkar et al. (2024); Hossain et al. (2021). Some pre-batch DP-adversarial alignments exist Thakkar et al. (2024), but systematic evaluation of post-batch combinations is lacking.

This work examines the interaction between evasion defenses and DP-based privacy training across four threat configurations: no attacks, only evasion attacks, only privacy attacks, and simultaneous evasion and privacy attacks. We use standard neural architectures and learning environments, analyzing outcomes empirically and through optimization dynamics and sensitivity control. Figure 1 summarizes the workflow.

The following research questions guide the investigation:

**RQ1**. How does the combination of defense methods affect performance on the primary learning task? In particular, does the degradation in accuracy exceed or fall below the degradation observed when each defense is applied independently?

**RQ2**. How does differential privacy influence susceptibility to evasion attacks in models protected by dedicated evasion defenses?

**RQ3**. How do evasion-oriented defenses affect the resistance of differentially private models to privacy attacks?

Our findings indicate that the interaction between defenses is not straightforward. In some configurations, combining methods yields complementary effects that enhance overall security characteristics without disproportionate loss of utility. In other cases, interactions introduce interference that weakens robustness or amplifies privacy risks, highlighting the limitations of treating defenses as modular components that can be stacked without careful analysis. These observations underscore the importance of evaluating machine learning systems under composite threat models rather than relying solely on single-threat benchmarks.

The main contributions of this work are summarized as follows:

- We provide a systematic empirical study of combined defenses against evasion and privacy attacks across composite threat scenarios, focusing on post-batch and hybrid robustness methods in combination with differential privacy.

- We examine how joint deployment affects primary-task performance, evasion robustness, and privacy leakage, highlighting that utility degradation is neither consistently additive nor subadditive, differential privacy can nullify standard evasion defenses, and adding robustness methods to DP models causes small but significant privacy leakage.

- We provide analytical insights that relate observed behaviors to the effects of gradient perturbations and regularization under DP. This analysis clarifies why interference and limited synergy occur.

- We emphasize the need for integrated evaluation frameworks that account for multi-factor threats in evasion and privacy-sensitive machine learning.

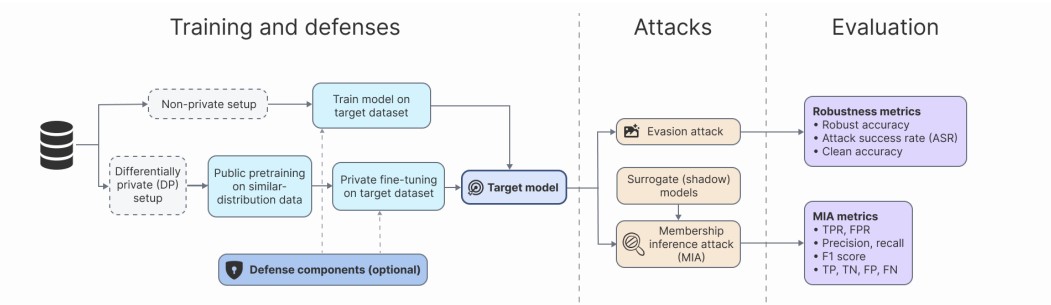

Figure 1: End-to-end experimental pipeline. We train a target classifier either in a standard non-private regime (top) or under differential privacy (DP) (bottom) via public pretraining on similar-distribution data followed by private fine-tuning on the target dataset. Evasion defenses can be enabled during training as optional components. The resulting target model is evaluated against two threat classes: an evasion attack and a membership inference attack that leverages surrogate (shadow) models.

These contributions provide a more complete view of the design of trustworthy machine learning systems, showing that effective protection requires coordinated consideration of multiple threat vectors rather than isolated optimization.

## 2 PROBLEM STATEMENT

Let $f : \mathbb{R}^d \to \Delta^K$ denote a neural network classifier mapping an input $x \in \mathbb{R}^d$ to a probability vector $f(x) \in \Delta^K$ over $K$ classes. The associated classification rule is defined as

$$h(f, x) = \arg \max_{i \in [1, \dots, K]} f(x)_i. \tag{1}$$

We introduce four formal definitions to characterize attacks and defenses, along with the evaluation metrics employed.

### 2.1 ATTACKS AND DEFENSES DEFINITIONS

**Definition 1** (Evasion Attack). *An* evasion attack *perturbs an input $x$ to produce $x'$ such that*

$$h(f, x') \neq h(f, x), \tag{2}$$

*and usually with additional perturbation constraint $\|x - x'\|_p \leq \delta$. The attack is evaluated using the* Attack Success Rate (ASR).

**Definition 2** (Membership Inference Attack). *A membership inference attack (MIA) is a binary classification task in which an adversary $\mathcal{A}$ predicts whether a sample $x$ belongs to the training dataset $\mathcal{D}$:*

$$\mathcal{A}(f, x) \in \{0, 1\}, \quad 1 \text{ indicates } x \in \mathcal{D}. \tag{3}$$

*The corresponding evaluation metric is* Advantage.

**Definition 3** (Robustness-Oriented Defense). *A robustness defense reduces susceptibility to evasion attacks. Let $\mathcal{B} = \{(x_i, y_i)\}_{i=1}^B$ be a training batch of size $B$, $\theta$ the model parameters, $\eta$ the learning rate, and $g_i = \nabla_\theta l(f, x_i, y_i)$ the gradient for sample $i$. We distinguish three classes:*

- ***Pre-batch defenses*** ($\mathcal{M}_{pre}$): *modify the batch before gradient computation*

$$\mathcal{B}' = \mathcal{M}_{pre}(\mathcal{B}), \tag{4}$$

  *e.g., evasion example generation or input augmentation Madry et al. (2017).*

- ***Post-batch defenses*** ($\mathcal{M}_{post}$): *act on gradients or after optimization*

$$g_i' = \mathcal{M}_{post}(g_i), \quad \theta \leftarrow \theta - \eta \frac{1}{B} \sum_{i=1}^B g_i', \tag{5}$$

*e.g., gradient regularization or clipping Ross & Doshi-Velez (2018).*

- **Hybrid defenses** ($\mathcal{M}_{hybrid}$): *combine pre- and post-batch operations*

$$\mathcal{B}' = \mathcal{M}_{pre}(\mathcal{B}), \quad g_i' = \mathcal{M}_{post}(\nabla_\theta l(f, x_i', y_i')), \quad \theta \leftarrow \theta - \eta \frac{1}{B} \sum_{i=1}^{B} g_i'. \tag{6}$$

*Example: TRADES Zhang et al. (2019).*

**Definition 4** (Differentially Private Model). *A differentially private (DP) model $f_{DP}$ is trained such that parameter updates satisfy $(\varepsilon, \delta)$-differential privacy. For each gradient*

$$\tilde{g}_i = g_i / \max(1, \frac{\|g_i\|_2}{C}) + \mathcal{N}(0, \sigma^2 C^2 I), \tag{7}$$

*where $C$ is the clipping norm and $\mathcal{N}(0, \sigma^2 C^2 I)$ is Gaussian noise. The update rule is*

$$\theta \leftarrow \theta - \eta \frac{1}{B} \sum_{i=1}^{B} \tilde{g}_i. \tag{8}$$

*This bounds the influence of individual samples, mitigating membership inference attacks.*

## 2.2 FORMAL PROBLEM

Given a classifier $f$, training dataset $\mathcal{D}$, robustness-oriented defense $\mathcal{M}_R \in \{\mathcal{M}_{\text{pre}}, \mathcal{M}_{\text{post}}, \mathcal{M}_{\text{hybrid}}\}$, and a DP mechanism $\mathcal{M}_P$ producing $f_{\text{DP}}$, the goal is to quantify the combined effect on:

1. Task performance on nominal inputs,
2. Susceptibility to evasion attacks ($T_R$),
3. Membership inference vulnerability ($T_P$).

Evaluation is conducted under four threat scenarios: no attacks, only evasion attacks, only privacy attacks, and simultaneous presence of both.

## 2.3 METRICS

### 2.3.1 ATTACK SUCCESS RATE (ASR)

For an evasion attack, ASR is defined as

$$\text{ASR} = \frac{1}{|\mathcal{D}_{\text{correct}}|} \sum_{x \in \mathcal{D}_{\text{correct}}} \mathbf{1}\{h(f, x') \neq h(f, x)\}, \tag{9}$$

where $\mathcal{D}_{\text{correct}}$ is the set of correctly classified images. An ASR of 1 indicates that all originally correct images are misclassified after the attack; lower values indicate stronger robustness.

### 2.3.2 MEMBERSHIP INFERENCE ADVANTAGE

Advantage is defined as

$$\text{Advantage} = \text{TPR} - \text{FPR}, \tag{10}$$

where TPR is the fraction of training samples correctly identified as members and FPR is the fraction of non-training samples misclassified as members.

**Rationale for using Advantage:** Differentially private training often limits the achievable FPR, making ROC-based metrics such as AUC or ROC-AUC unreliable for comparing models. In DP settings, it may be impossible to maintain identical FPR values for all models; hence, AUC or ROC curves can exaggerate or underestimate privacy leakage. Advantage provides a normalized, interpretable measure of membership leakage that reflects the distinguishability between training and non-training samples under practical DP constraints.

The metric advantage $\approx 1$ indicates maximal privacy leakage. An advantage of $\approx 0$ indicates negligible membership inference capability. An advantage exceeding 0.03–0.05 is typically considered a noticeable privacy leakage. An advantage exceeding 0.1 indicates a severe privacy breach.

These metrics are computed consistently across all threat scenarios, ensuring a representative evaluation of both robustness and privacy properties.

# 3 CONCEPTUAL ANALYSIS OF CONFLICTING EFFECTS OF DP AND GRADIENT DEFENSES

This section offers a conceptual discussion of the mechanisms behind the interaction between differential privacy (DP) and gradient-based evasion defenses. The discussion relies on simplified analytical analysis and aims to clarify why a conflict between these mechanisms can arise, rather than providing a rigorous mathematical derivation. A more detailed empirical investigation of this effect is presented in Section 4.

Gradient-based robust defenses operate by modifying the training dynamics to reduce model sensitivity to adversarial perturbations. In a simplified view, a single-step parameter update combining robust regularization with DP noise can be expressed as:

$$\theta_{t+1} = \theta_t - \eta\big(\nabla_\theta \mathcal{L}(\theta_t) + \sum_i \lambda_i \nabla_\theta R_i(\theta_t) + \epsilon_t\big), \quad \epsilon_t \sim \mathcal{N}(0, \sigma^2 I)$$

where $R_i(\theta)$ represents the gradient-based regularization term for the $i$-th defense (e.g., TRADES, LS, or IGR), and $\epsilon_t$ is the DP-induced noise. Each regularization term contributes a deterministic gradient that guides parameters toward regions of reduced sensitivity to specific attack directions.

DP noise introduces stochastic perturbations in the same parameter space. When the magnitude of $\epsilon_t$ is comparable to or larger than the combined robust gradients $\sum_i \lambda_i \nabla_\theta R_i$, the noise can displace parameters away from locally stabilized regions, potentially neutralizing the protective effect of the regularizations. This qualitative mechanism applies to all considered gradient-based defenses:

- TRADES: reduces sensitivity of input gradients for adversarial directions.

- Label Smoothing (LS): softens the output distribution, indirectly stabilizing input gradients.

- Input Gradient Regularization (IGR): directly penalizes large input gradients.

For an adversarial input $x + \delta$, a linearized loss illustrates how DP may affect effective gradients used by PGD attacks:

$$\nabla_x \mathcal{L}(x, \theta_T) \approx \nabla_x \mathcal{L}(x, \theta^*) + \nabla_x^2 \mathcal{L} \cdot (\theta_T - \theta^*),$$

where $\theta_T$ is the parameter vector after $T$ updates including DP noise. The second term represents a simplified estimate of the perturbation introduced by DP. When this term is comparable to the robust gradient contributions from $R_i$, the effective protection can be substantially reduced.

Consequently, the expected advantage against PGD attacks,

$$Adv_{PGD}(\theta_T) = Acc_{clean} - Acc_{PGD},$$

may decrease significantly if

$$\|\eta \epsilon_t\| \geq \Big\| \sum_i \lambda_i \nabla_\theta R_i \Big\|.$$

In this conceptual view, robustness gradients push parameters toward locally stable regions, while DP noise randomly perturbs them, creating a potential conflict.

Overall, these considerations indicate that naively combining differential privacy with multiple gradient-based evasion defenses may reduce robustness, and motivate the more detailed empirical study presented in Section 4.

## 4 EXPERIMENTS

### 4.1 METHODOLOGY

The objective is to determine whether privacy and evasion defenses interact when deployed together in a controlled multi-factor threat scenario. Given the limited literature on combined evasion and privacy settings, we analyze a concrete case where potential conflicts can be reliably observed, providing a basis for broader future studies. The overall workflow, including joint defense deployment, is shown in Figure 1.

To isolate defense effects, we construct a controlled benchmark with fixed dataset partitioning, architecture, training budget, and optimization; only defense components vary. This design attributes observed differences directly to defense combinations and enables diagnostic analysis.

We use CIFAR-10 Krizhevsky & Hinton (2009), reserving the official test set for reporting. The original training set is partitioned as follows: 25,000 images for target model training; 5,000 for validation and hyperparameter selection; 20,000 for training shadow models used in the OSLO membership inference attack Peng et al. (2024); and 10,000 for experimental testing. This split ensures adequate training capacity and statistical independence between shadow and target data, consistent with standard shadow-model assumptions. CIFAR-100 Krizhevsky & Hinton (2009) is used for public pretraining. Differentially private models are initialized from CIFAR-100 features and fine-tuned on the CIFAR-10 training split without access to private labels during pretraining.

Training images are augmented using random cropping with four-pixel padding and random horizontal flips, followed by normalization. Validation and test images are normalized without augmentation.

ResNet-18 He et al. (2016) serves as the backbone due to its widespread use in robustness and differential privacy research and its balance of capacity and efficiency on CIFAR-scale tasks.

Two training pipelines are evaluated. The first uses standard non-private training on CIFAR-10 as a utility and robustness baseline. The second applies differentially private training via DP-SGD Abadi et al. (2016), implemented as private fine-tuning on CIFAR-10 from a publicly pretrained CIFAR-100 initialization. Public pretraining mitigates utility degradation caused by DP noise and gradient clipping.

For non-private training, ResNet-18 is trained from scratch using SGD with momentum 0.9, weight decay $5 \times 10^{-4}$, learning rate 0.05, batch size 128, for 100 epochs. The final epoch model is selected. All experiments are repeated ten times. Detailed configurations of defenses and attacks are provided in the next subsection.

### 4.2 ATTACK AND DEFENSE HYPERPARAMETERS

To mitigate the accuracy degradation typically associated with training differentially private (DP) models from scratch, all DP models were initialized from a public pretraining phase on CIFAR-100. Pretraining was performed using stochastic gradient descent (SGD) with a learning rate of 0.1 for 100 epochs. For private fine-tuning on the CIFAR-10 training split, we employed DP-SGD with a maximum gradient clipping norm of $C = 1.0$, a learning rate of 0.05, and a logical batch size of 1024. Fine-tuning was conducted for 30 epochs to achieve a target privacy budget of $\varepsilon = 8.0$ at $\delta = 10^{-5}$. To balance privacy and model utility, a preliminary architectural evaluation compared fine-tuning only the classification head versus fine-tuning the last residual block. Fine-tuning the head alone yielded a clean accuracy of approximately 68.26% at $\varepsilon = 8.0$, whereas fine-tuning the entire last residual block improved accuracy to 79.42%. Based on these results, all DP-SGD updates were restricted to the last residual block in subsequent experiments. Experiments with tighter privacy bounds ($\varepsilon = 5.0$) showed substantial utility loss, precluding meaningful combination with evasion defenses.

We evaluated three defense methods against evasion attacks. Label Smoothing (LS) Müller et al. (2019), Input Gradient Regularization (IGR) Ross & Doshi-Velez (2018), and TRADES Zhang et al. (2019), both independently and in conjunction with DP-SGD. For LS, a hyperparameter sweep over $\alpha \in 0, 0.01, 0.03, 0.05, 0.1, 0.5, 0.7, 1$ was conducted to select the smoothing parameter. The sweep revealed a clear trade-off between standard accuracy and evasion robustness: hard labels ($\alpha = 0$)

achieved high clean accuracy (89.0%) but no robustness (ASR 89.4%, accuracy after attack 0.0%), whereas uniform labels ($\alpha = 1$) prevented learning (clean accuracy 9.2%). Moderate values such as $\alpha = 0.05$ slightly improved clean accuracy to 90.2% but provided limited robustness (ASR 83.6%). The optimal balance occurred at $\alpha = 0.5$. Clean accuracy remained 88.0%, ASR decreased to 75.0%, and accuracy after attack reached 15.2%. Higher smoothing ($\alpha = 0.7$) reduced all three metrics. Consequently, $\alpha = 0.5$ was fixed for all LS experiments.

TRADES was configured with a perturbation bound of $\varepsilon = 8/255$, 10 projected gradient descent steps of size 2/255, and a trade-off parameter $\beta = 6$. For IGR, the penalty coefficient $\lambda$ governs the trade-off between standard accuracy and gradient regularization. Preliminary validation showed that overly large values (e.g., $\lambda = 1.0$ or $\lambda = 0.05$) led to severe accuracy degradation under DP, whereas very small values failed to regularize effectively. We therefore selected $\lambda \in 0.01, 0.001$, which maintained stable trade-offs, achieving clean accuracies of 69.12% and 78.10% under DP and 83.32% and 87.95% in the non-private setting.

Robustness was evaluated using the Projected Gradient Descent (PGD) attack Madry et al. (2017) under the $L_\infty$ norm, with a maximum perturbation of $\varepsilon = 4/255$ and step size $\alpha = 1/255$, scaled according to the dataset normalization. The attack was executed for 10 steps with a batch size of 64, providing a consistent measure of model susceptibility to evasion perturbations.

To quantify privacy leakage, models were subjected to the OSLO membership inference attack Peng et al. (2024). This attack uses surrogate models and evasion perturbations to infer whether individual samples belong to the training set. The OSLO attack ensemble included 12 diverse surrogate models: one Inception, one ResNeXt-50, one PreActResNet-18, three independently trained DenseNet-121 models, three VGG-13 models, and three ShuffleNet models. Adversarial optimization within the attack was performed under an $L_\infty$ norm constraint of $\varepsilon = 80/255$ with step size 2/255 over 10 iterations, using 80 subprocedures. Hyperparameter sweeps over the decision threshold $\tau \in 0.3, 0.1, 0.03, 0.01, 0.003, 0.001, 0.0003, 10^{-4}, 3 \times 10^{-5}, 10^{-5}$ were conducted to select $\tau$ minimizing the False Positive Rate for each target model. The reported metric is the Membership Inference Advantage at this optimal threshold, ensuring a confident and comparable estimate of privacy leakage across all model configurations.

## 4.3 RESULTS OF EXPERIMENTS

We summarize the results under ten evaluation settings that directly correspond to the research questions formulated above. Setting (i), which accounts for the absence of attacks and evaluates primary-task utility, addresses RQ1 by quantifying the extent to which the joint application of defense methods influences clean accuracy relative to their independent application. Setting (ii), which evaluates robustness under PGD-based evasion attacks, addresses RQ2 by examining how differential privacy modifies susceptibility to evasion perturbations in models equipped with robustness-oriented defenses. Setting (iii), which evaluates resistance to OSLO membership inference attacks, addresses RQ3 by analyzing how evasion defenses influence privacy risk in differentially private models. For both PGD and OSLO, we report mean $\pm$ std over ten random seeds.

### 4.3.1 EFFECT OF COMBINED DEFENSES ON PRIMARY TASK PERFORMANCE (RQ1)

The first experiment addresses RQ1 by examining how the combination of defense methods affects performance on the primary learning task in the absence of attacks. Specifically, we evaluate whether the degradation in test accuracy under joint deployment exceeds or falls below the degradation observed when each defense is applied independently.

Table 1 reports the test accuracy for all configurations, including models trained without defenses, with evasion defenses only, with differential privacy only, and with their combinations. These results provide a direct view of absolute utility under each protection regime. To facilitate comparison across settings, Table 2 presents the corresponding accuracy changes in percentage points relative to the reference model trained without evasion defense and without differential privacy. This representation allows one to assess whether the utility loss induced by combining defenses aligns with, exceeds, or falls below the degradation associated with each component considered separately.

*Overall, the observed utility degradation under combined defenses does not follow a consistently additive or subadditive pattern. In some configurations, the joint effect is smaller than might be ex-*

Table 1: CIFAR-10 test accuracy (no attacks) under different combinations of evasion defenses and differential privacy (DP), including mean $\pm$ standard deviation over ten random seeds. Rows correspond to evasion defenses (including the absence of such defense), while columns indicate training without DP and with DP ($\varepsilon{=}8$). Each cell represents a specific combination of protection methods.

| Evasion defense | No DP | DP ($\varepsilon{=}8$) |
|---|---|---|
| None (Clean) | $87.73 \pm 1.08$ | $79.37 \pm 1.61$ |
| LS ($\alpha{=}0.5$) | $86.67 \pm 1.35$ | $78.81 \pm 0.69$ |
| IGR ($\lambda{=}0.001$) | $87.52 \pm 1.03$ | $77.55 \pm 0.98$ |
| IGR ($\lambda{=}0.01$) | $83.31 \pm 1.53$ | $68.62 \pm 1.47$ |
| TRADES | $73.30 \pm 0.49$ | $67.58 \pm 0.10$ |

Table 2: Accuracy degradation of defended models relative to the reference model without evasion defense and without differential privacy. All values are reported in percentage points. Boldface indicates a configuration in which the degradation under the combined application of evasion defense and differential privacy is statistically significantly smaller than the degradation induced by each component applied separately. Underlining indicates a configuration in which the combined degradation is statistically significantly larger than that of each method. These results demonstrate that the utility loss under joint deployment is neither additive nor consistently subadditive.

| Evasion defense | $\Delta$ No DP (pp) | $\Delta$ DP ($\varepsilon{=}8$) (pp) |
|---|---|---|
| None (baseline) | 0.00 | $-8.36$ |
| LS ($\alpha{=}0.5$) | $-1.06$ | $-8.92$ |
| IGR ($\lambda{=}0.001$) | $-0.21$ | $-10.18$ |
| IGR ($\lambda{=}0.01$) | $-4.42$ | $\underline{-19.11}$ |
| TRADES | $-14.43$ | **-20.15** |

*pected from the individual degradations, while in others it is substantially larger. This non-regular behavior indicates that simple composition rules are insufficient for predicting primary-task performance under multi-factor protection and highlights the need for further analysis before formulating practical guidelines for jointly robust and privacy-preserving model design.*

### 4.3.2 IMPACT OF DIFFERENTIAL PRIVACY ON SUSCEPTIBILITY TO EVASION ATTACKS (RQ2)

The second experiment addresses RQ2 by examining how enabling differential privacy influences susceptibility to evasion attacks in models protected by dedicated evasion defenses. To this end, we form matched pairs of models that implement the same evasion defense, trained either under standard non-private conditions or with DP-SGD at $\varepsilon{=}8$. Robustness is evaluated using the Attack Success Rate (ASR) under PGD attacks, with the corresponding clean accuracy reported for reference.

Table 3 summarizes the absolute clean accuracy and ASR for all configurations, including models trained without defenses, with evasion defenses only, with differential privacy only, and with their combinations. To facilitate direct comparison, Table 4 presents the pairwise differences (DP minus non-private) for matched defenses, highlighting the impact of differential privacy on both clean performance and evasion robustness.

*Overall, the introduction of differential privacy consistently reduces evasion robustness across all tested defenses. For TRADES and LS, which achieve substantial robustness gains in the non-private setting, DP versions exhibit a near-complete loss of protection against PGD attacks, with ASR approaching 100%. The clean accuracy also decreases under DP, though to a smaller extent. These observations indicate that differential privacy can substantially compromise robustness-oriented defenses, underscoring that naively combining DP with evasion defenses may not preserve evasion resilience and necessitate careful joint design and evaluation to ensure effectiveness.*

Table 3: Evasion robustness under PGD ($L_\infty$, $\varepsilon{=}4/255$, 10 steps). Top part: evasion defenses (LS/IGR/TRADES) without Differential Privacy (DP). Bottom part: DP models ($\varepsilon{=}8$) combined with the same evasion defenses. Robustness is read via Attack Success Rate (ASR, lower is better). The results indicate that combining DP with any of the considered evasion defenses substantially reduces robustness, effectively nullifying the protection these defenses provide against PGD attacks.

| Defense | Clean acc. (%) | ASR (%) |
|---|---|---|
| Clean | $87.73 \pm 1.08$ | $99.78 \pm 0.11$ |
| LS ($\alpha{=}0.5$) | $86.67 \pm 1.35$ | $61.08 \pm 1.52$ |
| IGR ($\lambda{=}0.01$) | $83.31 \pm 1.53$ | $83.19 \pm 0.07$ |
| IGR ($\lambda{=}0.001$) | $87.52 \pm 1.03$ | $95.23 \pm 0.14$ |
| TRADES | $73.30 \pm 0.49$ | $21.82 \pm 1.02$ |
| DP ($\varepsilon{=}8$) | $79.37 \pm 1.61$ | $99.88 \pm 0.00$ |
| DP + LS ($\alpha{=}0.5$) | $78.81 \pm 0.69$ | $99.47 \pm 0.15$ |
| DP + IGR ($\lambda{=}0.001$) | $77.55 \pm 0.98$ | $99.88 \pm 0.12$ |
| DP + IGR ($\lambda{=}0.01$) | $68.62 \pm 1.47$ | $99.66 \pm 0.47$ |
| DP + TRADES | $67.58 \pm 0.10$ | $98.46 \pm 0.27$ |

Table 4: Pairwise effect of Differential Privacy (DP) on evasion robustness (PGD attack) for matched defenses. Left part: models trained with only evasion defenses (LS/IGR/TRADES) and no Differential Privacy (DP). Right part: DP models ($\varepsilon{=}8$) combined with the same evasion defenses. Clean accuracy and Attack Success Rate (ASR) are reported.

| Matched pair (non-private $\to$ DP) | $\Delta$ clean acc. (pp) | $\Delta$ ASR (pp) |
|---|---|---|
| Clean $\to$ DP | $-8.36$ | $+0.10$ |
| LS $\to$ DP+LS | $-7.86$ | $+38.39$ |
| IGR (0.001) $\to$ DP+IGR (0.001) | $-9.97$ | $+4.65$ |
| IGR (0.01) $\to$ DP+IGR (0.01) | $-14.69$ | $+16.47$ |
| TRADES $\to$ DP+TRADES | $-6.72$ | $+76.64$ |

### 4.3.3 EFFECT OF EVASION DEFENSES ON THE PRIVACY OF DP MODELS (RQ3)

The third experiment addresses RQ3 by investigating whether adding robustness-oriented defenses (LS, IGR, TRADES) on top of differentially private training modifies the privacy leakage of models. We fix the privacy budget at $\varepsilon{=}8$ and compare the DP-SGD baseline to its combinations with the same evasion defenses. Privacy leakage is quantified using the OSLO membership inference attack, with Advantage (TPR$-$FPR; lower is better) as the primary metric, and auxiliary metrics reported for completeness.

Table 5: Membership inference attack results (OSLO). Top part: models trained with only evasion defenses (LS/IGR/TRADES) and no Differential Privacy (DP). Bottom part: DP models ($\varepsilon{=}8$) combined with the same evasion defenses. Advantage (primary, lower is better) is reported together with TPR/FPR. Main comparison is pairwise (e.g., LS vs DP+LS).

| Defense | Advantage | TPR | FPR |
|---|---|---|---|
| Clean | $0.063 \pm 0.051$ | $0.080 \pm 0.056$ | $0.017 \pm 0.006$ |
| LS ($\alpha{=}0.5$) | $0.057 \pm 0.015$ | $0.100 \pm 0.010$ | $0.043 \pm 0.006$ |
| IGR ($\lambda{=}0.01$) | $0.063 \pm 0.080$ | $0.557 \pm 0.081$ | $0.493 \pm 0.006$ |
| IGR ($\lambda{=}0.001$) | $0.057 \pm 0.046$ | $0.183 \pm 0.006$ | $0.127 \pm 0.040$ |
| TRADES | $0.160 \pm 0.056$ | $0.827 \pm 0.042$ | $0.667 \pm 0.015$ |
| DP-SGD ($\varepsilon{=}8$) | $0.003 \pm 0.042$ | $0.183 \pm 0.021$ | $0.180 \pm 0.036$ |
| DP-SGD + LS ($\alpha{=}0.5$) | $0.010 \pm 0.044$ | $0.207 \pm 0.067$ | $0.197 \pm 0.023$ |
| DP-SGD + IGR ($\lambda{=}0.001$) | $0.007 \pm 0.015$ | $0.197 \pm 0.032$ | $0.190 \pm 0.040$ |
| DP-SGD + IGR ($\lambda{=}0.01$) | $0.003 \pm 0.045$ | $0.227 \pm 0.021$ | $0.223 \pm 0.055$ |
| DP-SGD + TRADES | $0.010 \pm 0.070$ | $0.207 \pm 0.047$ | $0.197 \pm 0.045$ |

Table 6: Membership inference leakage (OSLO) for *differentially private* models ($\varepsilon$=8) and the same DP training augmented with *evasion defenses* (LS, IGR, TRADES). Advantage values are taken from Table 5; $\Delta$ denotes the change in mean Advantage relative to the DP-SGD baseline. Although the increases in leakage are relatively small in absolute terms, they are statistically significant, indicating that the formal guarantees of differential privacy for models combining multiple defenses cannot be taken based solely on the classical DP parameters when additional robustness methods are present.

| DP configuration ($\varepsilon$=8) | Advantage | $\Delta$ vs DP |
|---|---|---|
| DP | $0.003 \pm 0.042$ | 0.000 |
| DP + IGR ($\lambda$=0.01) | $0.003 \pm 0.045$ | 0.000 |
| DP + IGR ($\lambda$=0.001) | $0.007 \pm 0.015$ | +0.004 |
| DP + LS ($\alpha$=0.5) | $0.010 \pm 0.044$ | +0.007 |
| DP + TRADES | $0.010 \pm 0.070$ | +0.007 |

Table 5 reports the absolute leakage values for all configurations, including models trained without DP, with DP only, and with DP combined with evasion defenses. To simplify comparison across DP variants, Table 6 presents the corresponding mean Advantage and the changes relative to the DP-SGD baseline. This representation allows one to assess whether robustness-oriented defenses exacerbate, reduce, or leave the membership inference vulnerability in private models unchanged.

*Overall, adding evasion defenses to DP models results in only minor increases in privacy leakage, which are substantially smaller than the severe loss of evasion robustness observed in RQ2. Nevertheless, even these small increases are non-negligible, as the observed leakage varies depending on the combination of defenses. This indicates that the formal guarantees of differential privacy cannot be assumed in isolation based solely on the chosen privacy parameters, but should be considered in conjunction with other applied defenses. In practice, the effective privacy protection of a model may differ from nominal DP parameters when additional robustness methods are employed, highlighting the need for integrated evaluation of multi-defense systems.*

## 5    CONCLUSION AND FUTURE WORK

This study investigated how robustness to evasion attacks and differential privacy interact during neural network training, addressing three research questions: RQ1 on primary-task performance, RQ2 on evasion robustness under DP, and RQ3 on privacy leakage when evasion defenses are added.

The results show that combining defenses yields non-additive effects. For RQ1, the accuracy degradation under combined defenses cannot be consistently described as either additive or subadditive. For RQ2, differential privacy substantially reduces the effectiveness of standard evasion defenses, often removing their protective effect entirely. For RQ3, combining DP with evasion defenses produces small but statistically significant increases in privacy leakage, indicating that standard DP guarantees cannot be assumed when other defenses are present.

This work fills an important research gap: previous studies mostly considered defenses in isolation, leaving their interactions under multiple simultaneous threats unclear. Our empirical and analytical results show both interference and limited complementary effects, highlighting the need for evaluation methods that simultaneously account for multiple threats.

In future work, we will explore approaches to jointly manage accuracy, robustness, and privacy tradeoffs, design adaptive defenses against combined threats, and extend the analysis to other architectures, datasets, and attack types. The aim is to provide practical guidance for building machine learning systems that remain both robust and privacy-preserving in realistic evasion environments.

## ACKNOWLEDGMENTS

This work was supported by a grant provided by the Ministry of Economic Development of the Russian Federation in accordance with the subsidy agreement (agreement identifier

000000C313925P4G0002) and the agreement with the Ivannikov Institute for System Programming of the Russian Academy of Sciences dated June 20, 2025, No. 139-15-2025-011.

The results were obtained using the equipment of the Shared Research Facility, Shared Research Center of the Ivannikov Institute for System Programming of the Russian Academy of Sciences (SRC ISP RAS)

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
