# OpenReview forum: "Beyond Isolation: Effects of Defense Combinations on Evasion and Privacy Risks in Machine Learning"
_mathai.club/MathAI/2026/Conference — 2026 Oral_

### Official Review · Reviewer_BA8k · 2026-03-12
**This paper presents a diagnostic study of the problem of defense  interference in machine learning. However, it merely acknowledges the existence of a conflict.**

**Rating:** 4
**Confidence:** 4

**Review:**

The paper raises a highly relevant problem in modern machine learning (ML): the study of defense mechanisms under multi-component threats. In reality, ML systems must withstand not just one, but several attacks simultaneously (e.g., adversarial attacks during inference and data privacy attacks). However, the vast majority of existing works consider and optimize defenses in isolation.
The central thesis of the paper is that simple stacking of defenses developed separately does not guarantee overall security and can lead to unforeseen negative consequences: decreased accuracy, reduced robustness, or increased privacy data leakage.
The strict methodological foundation is an undeniable strength of the work. The paper introduces formal definitions for various types of defenses (pre-batch, post-batch, hybrid) and attacks (PGD for evasion, OSLO for membership inference). It considers four corner scenarios, allowing the isolation of interaction effects: (i) no attacks, (ii) evasion only, (iii) privacy only, (iv) both attacks simultaneously. Using a fixed dataset (CIFAR-10/CIFAR-100), architecture (ResNet-18), and hyperparameters allows asserting that the observed differences are caused specifically by the combination of defenses, rather than random factors. The authors justify using the Advantage metric (TPR - FPR) instead of AUC, noting that in DP models, FPR is often limited, making ROC curves incomparable.
However, the main limitation is the evaluation solely on the CIFAR-10 and CIFAR-100 datasets. These are relatively simple and small datasets (32x32 pixels, 10/100 classes), and the results might not transfer to more complex tasks (medical imaging, natural language processing), where defenses might interact differently.
Secondly, only the ResNet-18 architecture was investigated. It is unclear how combinations of defenses will behave on deeper networks, other architectures (DenseNet, EfficientNet), and especially on transformers, whose learning mechanisms are fundamentally different.
Thirdly, only a relatively high value of ∊=8 (weak privacy) is used. Even though accuracy drops catastrophically at ∊=5, the paper's conclusions remain valid only for a narrow range of weak defense.
Furthermore, the trade-off between accuracy and robustness is insufficiently explored. The drop in robustness could be a consequence of the overall decrease in accuracy due to differential privacy. Since differential privacy itself smooths the loss landscape (acting as a regularizer), it is possible that additional methods (LS, IGR, TRADES) are simply redundant—leading to a "saturation" of defense rather than a conflict of methods.
Finally, the presented research is purely diagnostic: it identifies the problem but does not propose ways to solve it.
This work is an diagnostic study that clearly formulates and visually demonstrates the problem of defense interference in machine learning. Its main strength lies in the formulation of the question and the purity of the experiment, proving that "security is not additive." However, its main weakness is the limited generalizability of the obtained results and the lack of deep analysis going beyond merely stating the fact of the conflict. The paper leaves open questions about the nature of this conflict on more complex data and architectures, its behavior at different privacy levels, and, most importantly, the ways to overcome it.

---

> ### Author Rebuttal · Authors · 2026-03-13
>
> We thank the reviewer for a careful analysis of the work and for the substantive comments. In response to the remarks, we would like to clarify several points.
>
> Regarding “These are relatively simple and small datasets… only the ResNet-18 architecture was investigated,” we agree that using CIFAR and a single architecture does not ensure generalization to all tasks. However, the study’s goal was to demonstrate that defense methods can interact non-additively, not to generalize universally. A single valid example suffices to show the existence of such a conflict. The experimental design and controlled conditions align with this objective, especially given the lack of prior research on the additivity of defenses for these attack classes.
>
> Concerning the comment “.. only a relatively high value of ∊=8 (weak privacy) is used”, we note that the value ∊=8 was optimal in terms of the balance between protection and accuracy within the conducted experiments, as detailed in Section 4.2. The Advantage metric for MI attacks decreases over 20-fold to 0.003, approximating random guessing, confirming that this privacy level suffices to demonstrate method interactions without compromising conclusions regarding conflict.
>
> Regarding the remark “Since differential privacy itself smooths the loss landscape (acting as a regularizer), it is possible that additional methods .. may be simply redundant -- leading to a ‘saturation’ of defense rather than a conflict of methods”, our observations do not support the “saturation” hypothesis. Table 3 shows evasion attack success on DP models (99.88%) is comparable to unprotected models (99.78%), indicating negligible DP effect on evasion robustness. ASR was computed only for correctly classified instances, avoiding distortion and permitting accurate assessment of defense interactions.
>
> Finally, regarding the comment “the presented research is purely diagnostic”, we confirm that the work is diagnostic in nature and focuses on studying the problem rather than solving it. A formal description of the conflict between defense methods is necessary for the subsequent search for solutions. The precise identification and visualization of a previously unreported problem constitute an independent scientific contribution and highlight the study’s significance, even within a limited experimental context.
>
> We hope these clarifications clarify the objectives and justification of the experiments and demonstrate the correctness of the work conclusions.

---

### Official Review · Reviewer_dpas · 2026-03-12
**The paper presents practical results showing that a naive combination of defense mechanisms does not always lead to positive effects**

**Rating:** 7
**Confidence:** 3

**Review:**

The authors investigate how various combinations of defense mechanisms influence the robustness and confidentiality of data in machine learning systems.

Four key threat scenarios are clearly identified and analyzed: absence of attacks, attacks bypassing protection only, attacks on confidentiality only, and simultaneous attacks that bypass protection and target confidentiality.

A high-quality literature review has been conducted, with numerous references to up-to-date articles that address only individual aspects of trust in machine learning.

The work provides a rigorous mathematical description of the problem, attack models, defense mechanisms, and the metrics used.

The authors present both solid theoretical justifications and detailed empirical experiments along with their analysis. The main practical takeaway is that a naïve combination of all available defenses does not always produce positive results.

In conclusion, the work combines theoretical analysis, rigorous mathematics, an extensive literature review, and practical experiments with valuable conclusions.

---

> ### Author Rebuttal · Authors · 2026-03-13
>
> We thank the reviewer for a careful analysis of the work.

---

### Official Review · Reviewer_SdEQ · 2026-03-13

**Rating:** 7
**Confidence:** 3

**Review:**

This paper presents an empirical investigation into the intersection of machine learning security and privacy. Specifically, it examines how combining Differential Privacy with gradient-based evasion defenses affects model utility and robustness against adversarial evasion. Through experiments on CIFAR-10 using ResNet-18, the authors demonstrate that naively stacking these defenses leads to non-trivial interference. Authors report that applying DP effectively nullifies the protective benefits of standard evasion defenses, while the addition of evasion defenses to DP models causes a slight but statistically significant increase in privacy leakage.

Comments:
1) The empirical evaluation is restricted to a single dataset, single architecture and a single type of evasion attack. While sufficient to prove the existence of defense interference, evaluating larger datasets or alternative attacks would make the claims much stronger

2) The paper does an excellent job of diagnosing a critical defense interference but does not propose a novel architectural or algorithmic solution to resolve the conflict.

---

> ### Author Rebuttal · Authors · 2026-03-13
>
> We thank the reviewer for the careful analysis and substantive comments.
>
> "The paper does an excellent job of diagnosing a critical defense interference but does not propose a novel architectural or algorithmic solution to resolve the conflict," we confirm that our work is diagnostic in nature. Its main focus is on studying the problem rather than providing a solution. A formal description of the conflict between defense methods is necessary before attempting potential solutions; this was our goal.
>
> We plan to continue this research and aim to develop the mathematical ideas described in Section 3. Future work may propose both mathematical and applied solutions. At this stage, our priority was to study interactions between different methods and provide an initial formal description of the problem.
>
> This also addresses the comment "While sufficient to prove the existence of defense interference, evaluating larger datasets or alternative attacks would make the claims much stronger." We agree that broader evaluations would strengthen the claims. However, we considered it more important to explore additional research questions related to the conflict rather than expand datasets and models in the current work. When proposing solutions in future studies, we will consider a significantly larger set of models and datasets. This work lays the foundation for such subsequent research.

---

### Decision · Program_Chairs · 2026-03-14

**Decision:**

Accept (Oral)

**Comment:**

Dear Author(s),

On behalf of the Program Committee of the International Conference on Mathematics of Artificial Intelligence (MathAI 2026), we are pleased to inform you that your paper has been accepted for an oral presentation at MathAI 2026.

Your paper was evaluated through a rigorous two-stage review process involving both automated screening and expert review by members of the Program Committee. The reviewers recognized the quality and contribution of your work.

Presentation details:

- Format: Oral presentation (15–20 minutes + 5 minutes Q&A)
- Mode: You may present either in person (offline) at the conference venue in Sirius, Russia, or remotely via Zoom. Please indicate your preferred mode when confirming your participation.
- Conference dates: Marh 30 - April 3, 2026
- Website: https://mathai.club

Next steps:

1. Please confirm your participation and presentation mode by replying to this email mathai.club@yandex.ru no later than March 15, 2026 18:00 Moscow time.
2. If you plan to attend in person, the organizing committee will provide accommodation details separately.
3. Please prepare your final camera-ready manuscript according to the formatting guidelines available at https://mathai.club and upload it to OpenReview by March 15, 2026 18:00 Moscow time.

Should you have any questions regarding the program, logistics, or your presentation slot, please do not hesitate to contact us.

We look forward to your contribution to MathAI 2026.

With kind regards,

MathAI 2026 Program Committee
International Conference on Mathematics of Artificial Intelligence
https://mathai.club
OpenReview: https://openreview.net/group?id=mathai.club/MathAI/2026/Conference
Telegram: https://t.me/MathAI_club
Email: mathai.club@yandex.ru